# A Detailed Insight into Acoustic Attenuation in a Static Bed of Hydrophilic Nanosilica

**DOI:** 10.3390/nano12091509

**Published:** 2022-04-28

**Authors:** Syed Sadiq Ali, Agus Arsad, SK Safdar Hossain, Mohammad Asif

**Affiliations:** 1Department of Chemical Engineering, King Faisal University, P.O. Box 380, Al-Ahsa 31982, Saudi Arabia; ssali@kfu.edu.sa (S.S.A.); snooruddin@kfu.edu.sa (S.S.H.); 2UTM-MPRC Institute for Oil and Gas, School of Chemical and Energy Engg, Faculty of Engineering, Universiti Teknologi Malaysia, Johor Bahru 81310, Malaysia; agus@utm.my; 3Department of Chemical Engineering, King Saud University, P.O. Box 800, Riyadh 11421, Saudi Arabia

**Keywords:** acoustic vibrations, nanosilica, frequency analysis, attenuation, freeboard region, granular bed

## Abstract

The commercial utilization of bulk nanosilica is widespread in concrete, rubber and plastics, cosmetics and agriculture-related applications, and the market of this product is projected to exceed USD 5 billion by 2025. In this investigation, the local dynamics of a nanosilica bed, excited with sinusoidal acoustic waves of different frequencies, were carefully monitored using sensitive pressure transducers to obtain detailed insights into the effectiveness of sound waves as a means of energy transport inside the bed. The evolution of wave patterns and their frequency and power distributions were examined both in the freeboard and in the static bed. These results were compared with those obtained by using an empty column. The acoustic frequency strongly affected the signal power. The average power of the acoustic signal in the freeboard region was twice higher than that for the empty column, whereas the same (power) ratio decreased to approximately 0.03 inside the bed for 300 Hz. However, at 360 Hz, the power ratio was substantially lower at 0.24 and 0.002 for the freeboard and the granular bed, respectively, thereby indicating tremendous attenuation of acoustic waves in the granular media at all frequencies.

## 1. Introduction

Ultrafine particles are widely used in process industries and laboratories owing to their tremendously large surface area, which enhances the surface-based rate processes. However, processing these particles is often challenging due to strong interparticle forces (IPFs), which lead to the formation of large agglomerates and compromise the effectiveness of these particles in actual applications. During the two-phase gas-solid processing, agglomeration may lead to poor interphase mixing, low heat, and mass transfer rates, thereby compromising the efficiency of the overall process [1,2,3,4,5].

Fluidization technology can be used to promote efficient interphase mixing between the gas phase and the resident solid phase of the bed. This technique improves the heat and mass transfer rates and lowers the energy consumption by limiting the pressure drop to the effective weight of the bed [6,7,8,9,10]. However, the physical properties of the solid phase have an important bearing on the nature of the fluidization. The fluidization of fine and ultrafine particles is particularly challenging due to the presence of strong IPFs, which leads to an uneven and non-homogeneous fluidization. To counteract the effect of IPFs, fluidization assistance is often employed by inputting additional energy. This approach helps improve the fluidization hydrodynamics by lowering the minimum fluidization velocity (U_mf_), increasing the bed expansion and suppressing the hysteresis phenomenon caused by the bed’s non-homogeneities [11,12,13,14,15,16].

One of the most widely reported assisted fluidization techniques in the literature is acoustic vibration, where sound waves act as a source of additional energy needed to overcome the IPFs [17,18]. In fact, the application of acoustics, especially nanoacoutics, in different areas of science, engineering, and medicine is a growing area of research [19]. Many studies have investigated the effect of the acoustic frequency and amplitude on fluidization hydrodynamics by monitoring the minimum fluidization velocity in the presence of sound waves. For example, Zhu et al. [20] varied the sound pressure level (SPL) and frequency in a short fluidized bed of hydrophobic nanoparticles and found that the acoustic field helped lower the U_mf_ and improve the bed homogeneity when the amplitude exceeded 95 dB. They detected a stronger effect of the sound at a lower frequency and higher amplitude. However, at frequencies greater than 2000 Hz, the effect of acoustics was reportedly negligible. Apart from lowering the minimum fluidization velocity, sound assistance can also improve the stability of the fluidized bed by suppressing the density fluctuations, thereby promoting segregation based on the density difference [21]. The improved hydrodynamics of sound–assisted fluidized beds can enhance the carbon capture efficacy of activated carbon, and impart greater homogeneity to a bed of binary nano-powders during their fluidization [22,23]. Similarly, the use of acoustics was reported to enhance the fluidization quality and drying rate of lignite, thereby resulting in a better product quality [24].

Xu et al. [25] studied the effect of acoustic waves on the fluidization of Geldart groups C and A particles in a shallow fluidized bed (height: 4–5 cm) by placing the sound source below the distributor. For both groups of particles, sound waves of 120 Hz frequency were most effective in lowering the U_mf_, whereas the theoretical resonant frequency was predicted to be 81 Hz. The authors also measured the SPL above the bed. However, at a fixed amplitude and frequency, the SPL values mostly decreased with the velocity increase for the group C particles. Meanwhile, SPL values recorded above the bed were relatively unaffected for group A particles above 1.5 cm/s. The authors argued that this behavior can be used to distinguish groups A and C particles. 

Herrera et al. [26] examined the formation of standing waves in fluidized beds of fine particles as a function of frequency by placing a sound source at the top of a 0.9 m long column and by measuring the SPLs at different positions along the bed height. The SPL profiles along the height showed good agreement with the classical one-dimensional wave model that accounted for the attenuation phenomenon due to the damping of acoustic vibrations in the fluidized bed. By using shallow beds (height: 4.5–8.3 cm), they obtained velocity of sound as low as 24 m/s for a 40-m fly ash fluidized above the minimum bubbling velocity. Kumar et al. [27] investigated the acoustic attenuation characteristics of a 1.9 m long fixed bed with different packings. By conducting the nonlinear regression of experimental data and using the predictions of the classical wave model, they calculated the attenuation coefficient and acoustic velocity in the bed and found that the attenuation increased along with frequency and decreased along with the increasing bed void fraction. Increasing in the void fraction also increased the acoustic velocity and natural frequencies. 

Cherntongchai et al. [28] determined the U_mf_ of a sound-assisted shallow fluidized bed (height ≈ 8.0 cm) of group A particles in the frequency range 50 Hz–500 Hz using a sound source located at the top of a long fluidization column with SPL fixed at 80 dB. By assuming that the SPL at the top of the bed is same as that of the sound source located at the top of the fluidization column, they predicted the formation of standing waves at different frequencies used in their experiments following the approach of Herrera et al. [26]. However, they did not consider the sound attenuation in the fluidized bed. They attributed the U_mf_ decrease at 50 Hz to the formation of a standing wave with kL=π/2, where *k* was the wave number, and *L* is the height of the fluidized bed. 

Al-Ghurabi et al. [29] monitored the pressure transients at different locations along the height of a 370-mm long sound-assisted fluidized bed of hydrophilic particles with strong agglomeration behavior. They found that the pressure fluctuations at the resonant frequency were significantly higher than those obtained at non-resonant frequencies, whereas the mean values of the pressure drop were mostly unaffected by the frequency. Surprisingly, the sound waves at frequencies significantly different from the resonant frequency hardly affected the bed hydrodynamics. Moreover, the gas velocity mitigated the effect of acoustic vibrations as the vigorous solid motion at high velocities absorbed the momentum of the sound waves. During their downward propagation, the sound waves undergo significant attenuation since the acoustic intensity, characterized by pressure fluctuations, in the lower bed region was significantly smaller than that in the upper region.

Among assisted fluidization techniques, sound-assisted fluidization has been widely investigated owing to its potential application as a low-cost alternative to the energy-intensive assisted fluidization technique of mechanical vibration. While most studies have mainly focused on improving fluidization hydrodynamics as discussed above, a deeper insight into the evolution of the wave structures and the strength of the acoustic field in the free board region and the granular bed has been clearly lacking in the literature. This study sets out to fill this gap by investigating the effect of sound waves along the height of a granular bed of highly porous nanosilica at different frequencies. The sound velocity, while being approximately 343 m/s in the air (and therefore in the free-board region with a negligible particle concentration) can be as low as 20 m/s in the granular bed [26]. The presence of these two acoustically interacting yet physically separate regions would lead to the evolution of different wave patterns given that the natural frequencies of standing waves are critically dependent upon the sound velocity. Unlike in previous studies where the sound effects were mainly characterized by fluidization behavior (e.g., minimum fluidization velocity), this work directly monitors the acoustic waves (with frequencies ranging from 200 Hz to 400 Hz emitted from the source) in the bed and the freeboard region by using sensitive pressure transducers. The highest acoustic frequency (i.e., 400 Hz) used was below the Nyquist frequency of 500 Hz. The experimental results were also compared with that of an empty column to clearly delineate the effect of the granular bed on the evolution and attenuation of wave structures. Such a study that clearly sheds light on the limitation of the sound waves as means of the fluidization assistance has in fact been long overdue. Results of this investigated clearly reveal that the effect of acoustic vibrations is mainly limited to the upper part of the granular bed. Even at a depth of 0.5 m from the upper bed interface with a sound source of 125 dB, the signal attenuation becomes too significant that any perceived effect of sound waves on the fluidized bed hydrodynamics hardly appears justifiable.

## 2. Experimental

### 2.1. Experimental Setup

The detailed schematic of the experimental setup is reported in Figure 1. A 1.5 m tall transparent perspex column with 0.07 m internal diameter (ID) was used as the test section. A 0.3 m long plenum was used as the calming section. A distributor, with 2 mm holes and 2.7% opening, was used to separate the test section from the calming section. The distributor was covered with a nylon mesh of pore size 20 μm to prevent the solid particles from passing through the holes of the distributor. A 0.3 m long and 0.15 m ID disengagement section was placed at the top of the test-section to minimize the entrainment losses. Acoustic vibrations were generated using a downward-facing loudspeaker placed above the disengagement section at a distance of approximately 1.75 m from the distributor. The loudspeaker was connected to the audio port of a laptop through an amplifier. A MATLAB program (MathWorks, Natick, MA, US) was used to generate sinusoidal waveform of the required frequency.

The acoustic waves were monitored using 10 different pressure transducers as shown in Figure 1. The positions of the pressure ports from the distributor are presented in Table 1, where the distances from the distributor (x) are normalized with respect to the distance of the sound source (i.e., L = 1.75 m) to yield dimensionless distance, $\bar{x}$, from the distributor. Highly sensitive bidirectional differential pressure transducers (PX163-005BD5V, Omega, Norwalk, CT, US) with a response time of 1 ms and a bandwidth of 1 kHz were used. The range of these transduces were ± 5 in H_2_O (i.e., ±1244 Pa) to ensure an accurate measurement of low-pressure events taking place in our acoustically perturbed system. The pressure transducers were connected to a data acquisition (DAQ) system (USB-6289, National Instruments, Austin, TX, US), operated using the LabVIEW software (National Instruments, Austin, TX, US). The pressure transient data were acquired at a rate of 1000 Hz.

### 2.2. Nanosilica Powder

Hydrophilic nanosilica (Aerosil 200, Evonik Industries AG, Essen, Germany) with a reported primary size of 12 nm and specific surface of 200 m^2^/g were used in our experiments. In a powdered bed, these materials have a porosity of approximately 0.977 and a tapped density of approximately 50 kg/m^3^. In dry dispersion, the nanosilica mainly takes the form as multi-level agglomerates with a wide size distribution varying from 2 μm to 100 μm [30,31]. 

### 2.3. Measurements and Calibration

The pressure fluctuations can be characterized as the root mean square of sound pressure, $p_{rms}$, which is the standard deviation of the recorded pressure transients or root mean square when the average pressure is zero
(1)$$p_{rms}=\sqrt{\frac{1}{\left(N-1\right)}\overset{N}{\underset{i=1}{\sum }}{\left(P_{i}-\bar{P}\right)}^{2}}$$
where *N* is the number of data points, $p_{i}$ is the instantaneous pressure in Pa, and $\bar{P}$ is its average value.

$p_{rms}$ can be converted into SPL (dB) as
(2)$$\mathrm{SPL}=20\log_{10}\left[\frac{p_{rms}}{P_{ref}}\right]$$
where $p_{ref}=2\times 10^{-5}$ Pa is the reference pressure. However, to obtain precise SPL values in the experiments, a calibration was performed to interpolate SPL with $p_{rms}$ using a Bruel and Kjaer, Type 4231 (Nærum, Denmark) sound pressure level meter. To this end, the pressure and SPL were recorded simultaneously by plugging both the SPL meter and pressure traducers in the same pressure tap. As shown in Figure 2, the pressure transducer output in terms of Pa can be related to SPL (dB) as
(3)$$\mathrm{SPL}=9.646ln\left(p_{rms}\right)+83.05$$

### 2.4. Methodology

The sound pressure level of the acoustic source (i.e., loud-speaker) was fixed at 125 dB. Two sets of experiments were carried out by varying the sound frequency and recording the pressure dynamics at 1000 Hz. An empty column was used in the first set of experiments, whereas a nanosilica bed was employed in the second set of experiments. The height of the bed was set to 0.78 m in the experiments. The experiments were carried out by varying frequencies in the range of 200 Hz to 400 Hz (i.e., 200, 220, 240, 270, 300, 320, 340, 360, 380, and 400 Hz).

## 3. Mathematical Model

### 3.1. Empty Bed

When an acoustic wave travels from one medium to another, a part of incident wave is reflected by the interface, whereas the other portion transmits though the interface to another medium. When the reflected wave overlaps the incident wave, a new waveform called standing waves is generated due to the superimposition of incident and reflected waves, which is termed as standing wave. The pressures of the reflected and transmitted acoustic waves at the interface are related with that of the incident wave in terms of the impedances of these two media as follows [32]:(4)$$\frac{p_{r}}{p_{i}}=\frac{z_{2}-z_{1}}{z_{2}+z_{1}}$$
(5)$$\frac{p_{t}}{p_{i}}=2\frac{z_{2}}{z_{2}+z_{1}}$$
where $p_{i}$ is the pressure of the incident wave, $p_{r}$ and $p_{t}$ are the pressures of the reflected and transmitted waves, respectively. The $z_{1}$ and $z_{2}$ are the specific acoustic impedances of the first medium and second media.

The one dimensional propagation of plane acoustic pressure waves in air can be described as [26,32]:(6)$$\frac{\partial^{2}p}{\partial t^{2}}-c^{2}\frac{\partial^{2}p}{\partial x^{2}}=0$$
where *c* is the speed of the acoustic wave in the medium, and *t* and *x* are the temporal and spatial coordinates, respectively. The longitudinal sound waves travel from the source towards the distributor, which acts as a closed boundary. The reflected wave from the distributor forms a standing wave when superimposing the incident wave. The general solution of Equation (6) is given as [32]
(7)$$p\left(x,t\right)=\sqrt{{\left(A+B\right)}^{2}{cos}^{2}\left(kx\right)+{\left(A-B\right)}^{2}{sin}^{2}\left(kx\right)}\;\;cos\left(\omega t-\phi \right)$$
where
(8)$$\phi ={tan}^{-1}\left(\frac{\left(A-B\right)}{\left(A+B\right)}tan\left(kx\right)\right)$$

Here, $k=\frac{\omega }{c}=\frac{2\pi f}{c}=\frac{2\pi }{\lambda }$, is the wave number, f is the frequency, ω is the angular frequency, and λ is the wavelength.

For the case of an empty column, the boundary is the distributor, which is a rigid thick perspex plate with a specific acoustic impedance of $3.2\times 10^{6}\;\mathrm{Pa}\cdot m^{-1}s$ [33]. Given that the specific acoustic impedance of air is $415\;\mathrm{Pa}\cdot m^{-1}s$, we have $z_{2}\gg z_{1}$, and hence $p_{r}\approx p_{i}$ in Equation (4). The amplitudes of the incident and reflected waves are equal. Therefore,
(9)$$p\left(x,t\right)=P_{0}\sqrt{\frac{{cos}^{2}\left(kx\right)}{{cos}^{2}\left(kL\right)}}\;\;cos\left(\omega t\right)$$
where $P_{0}$ is the amplitude of the waves originating from the source of acoustic vibrations (i.e., loudspeaker). 

The standing wave formation results in the development of nodes and antinodes in the closed column that can be predicted as follows using amplitude term in Equation (9)
(10)$$kL=\left(2n-1\right)\frac{\pi }{2};\;\;\;\;\;\;\;\;\;\;n=1,\;2,\;3,\ldots \ldots$$
which corresponds to the pressure antinodes or displacement nodes. Therefore, kL=π/2 gives rise to the first harmonic with one node and one antinode. For the case with *L*= 1.75 m and 27 °C, the fundamental frequency is approximately 50 Hz. Note that closed pipe will allow only odd harmonics. Therefore, the frequency of the third, fifth, and seventh harmonics will be 149, 249, and 348 Hz, respectively, given that the velocity of sound at 27 °C is approximately 348 m/s.

### 3.2. Static Granular Bed of Ultrafine Particles

During the propagation of acoustic waves through the freeboard region, part of the pressure intensity transverses through the interface and contributes to the augmentation of energy in the granular bed, whereas the remaining energy is reflected as shown in Figure 3. The upper layer of the bed acts as the first interface for the reflection and refraction of the acoustic waves. The transmitted wave $p_{t}{}_{1}$ travels through the bed and reaches the distributor at the bottom, which acts as the second interface. The reflected wave $p_{r}{}_{1}$ from the first interface superimposes with the incident wave $p_{i}{}_{1}$ to form a standing wave in the freeboard. However, the resultant standing wave in the freeboard is different from the standing waves formed in the completely empty column due to the difference between the two media. The fraction of sound intensity that is transmitted to the other medium and the one that is reflected back can be calculated by using the acoustic power transmission coefficient, $\alpha_{t}$ and acoustic power reflection coefficient, $\alpha_{r}$ [32].
(11)$$\alpha_{t}=\frac{I_{t}}{I_{i}}=\frac{p_{t}{}^{2}z_{1}}{p_{i}{}^{2}z_{2}}$$
(12)$$\alpha_{r}=\frac{I_{r}}{I_{i}}=\frac{p_{r}{}^{2}}{p_{i}{}^{2}}$$

Substituting Equations (3) and (4) into Equations (11) and (12) yields
(13)$$\alpha_{t}=\frac{4z_{1}z_{2}}{{\left(z_{1}+z_{2}\right)}^{2}}$$
(14)$$\alpha_{r}=\frac{{\left(z_{2}-z_{1}\right)}^{2}}{{\left(z_{1}+z_{2}\right)}^{2}}$$

The specific acoustic impedance can be calculated as [34]
(15)$$z=\rho_{b}c$$

The velocity of sound in a medium is calculated as [35,36]
(16)$$c=\frac{1}{\sqrt{\rho_{b}\left(\frac{\varepsilon }{K_{g}}+\frac{\left(1-\varepsilon \right)}{K_{p}}\right)}}$$
where $K_{g}$ and $K_{p}$ are the bulk moduli of the gas and solid particles in the bed, respectively. For air, $K_{g}=1.01\times 10^{5}\;\mathrm{Pa}$, and $K_{P}=3.67\times 10^{7}\;\mathrm{Pa}$ for SiO_2_. The speed of sound in the granular bed is therefore 45.4 m/s, whereas specific values of impedance are $2152.43{\;\mathrm{Pa}\;m}^{-1}s$ and $415{\;\mathrm{Pa}\;m}^{-1}s$ for the bed and air, respectively. Equations (13) and (14) are then used to calculate the acoustic power transmission coefficient, $\alpha_{t}=0.54$ and acoustic power reflection coefficient, $\alpha_{r}=0.46$. However, when the acoustic wave reaches the distributor, complete reflection takes place at the rigid distributor interface $\left(z_{2}=3.2\times 10^{6}{\;\mathrm{Pa}\;m}^{-1}s\right)$, leading to $\alpha_{r}\approx 1$ in Equation (14), given that $z_{2}\gg z_{1}$. Moreover, in the particles bed, we have $p_{t}{}_{1}>p_{r}{}_{2}$ due to the attenuation of acoustic resulting from the energy absorption by the nanosilica bed.

## 4. Results and Discussion

This section initially presents the experimental data for the empty column for different sound frequencies recorded at several locations along the height of the closed column shown in Figure 1. The case of the highly porous static bed of ultrafine nanosilica is considered afterwards. Unlike previous studies that were mostly carried out using shallow beds, using sensitive pressure transducers in this work allowed utilization of a relatively long bed with a height of 0.78 m. The sound amplitude and recoding locations were kept the same as the ones for the empty column to examine the evolution and attenuation of the wave patterns during their downward propagation through the nanosilica particle bed toward the distributor.

### 4.1. Empty Column Dynamics under Acoustic Vibrations

Although the experiments were carried out for 10 different frequencies ranging from 200 Hz to 400 Hz, Figure 4 shows the experimental data only for three selected acoustic frequencies (i.e., 220, 240 and 360 Hz) to examine how the difference in frequencies, whether small or large, affects the evolution of wave patterns in the empty column. The normalized distance of the measurement point from the distributor is shown as the boxed legend. Although the difference between 220 Hz and 240 Hz was not substantial, yet their wave dynamics significantly differed. For instance, the 240 Hz acoustics, demonstrated localized high-density pattern, resulting from the presence of different waves with close frequencies. Interestingly, the location closest to the sound source (i.e., $\bar{x}$ = 0.86) in Figure 4a,b (at 220 Hz and 240 Hz) had much smaller fluctuations compared with farther away location at $\bar{x}$ = 0.74. The amplitude of acoustic waves subsequently decreased while moving away from the source until reaching $\bar{x}$ = 0.40 (Figure 4a,b), which was slightly below the midpoint of the empty column. However, the wave amplitude demonstrated a modest increase farther away from the acoustic source (i.e., $\bar{x}$ = 0.29) before decreasing at $\bar{x}$ = 0.17 and progressively increasing in the lower portion of the closed empty column. The highest degree of disturbance was observed slightly above the distributor at $\bar{x}$ = 0.06, whereas a substantial attenuation in magnitude was observed immediately above the distributor. The difference in sound frequency, albeit insignificant, critically influenced the acoustic amplitude. For instance, at the same locations (e.g., at $\bar{x}$ = 0.06) the amplitude of acoustic disturbance for 240 Hz was almost twice as large as that for 220 Hz.

When the frequency increased to 360 Hz (Figure 4c), the fluctuations mostly showed a significant increase compared to those observed at the corresponding locations for 220 Hz and 240 Hz. For instance, at $\bar{x}$ = 0.06, the amplitudes of pressure fluctuations were as high as ±1000 Pa. By contrast, the fluctuations were substantially mitigated at other locations (e.g., $\bar{x}$ = 0.40, 0.17, and 0.01). The dependence of the wave amplitude on the location was mainly due to the superposition of the incidence wave with the one reflected from the distributor, which resulted in the formation of standing waves. Theoretically, there should have been a displacement node (or pressure antinode) at the distributor that acts as a closed boundary. On the contrary, the amplitude at this location was smaller than that at $\bar{x}$ = 0.06, which could be an artefact of the perforations present on the distributor. According to the wave model, when a closed empty column is subjected to acoustics of 360 Hz (i.e., kL ≅7.2π/2), which is close to the seventh harmonic (i.e., the third overtone) a standing wave with four nodes and an equal number of antinodes is formed.

The pressure transients in Figure 4 comprise of waves of different frequencies even though an acoustic wave of a single frequency was introduced in the column. This aspect of the acoustics dynamics is explored to obtain deep insights into this phenomenon by carefully analyzing the signals in the frequency domain. For the case of the 220 Hz acoustics shown in Figure 5a, two small-amplitude yet prominent peaks of comparable magnitude corresponding to 220 Hz and 320 Hz were observed at $\bar{x}$ = 0.86. Interestingly, peak of 320 Hz was more prominent than that of 220 Hz. Therefore, the total standard deviation of the acoustic fluctuations at $\bar{x}$ = 0.74 was 175 Pa, followed by 160 Pa at $\bar{x}$ = 0.06, where several acoustic frequencies were present, apart from the prominent peak at 220 Hz. The high amplitude waves at widely varying locations inside the column indicates the presence of pressure antinodes. Meanwhile, the lowest standard deviation of 25 Pa was recorded at $\bar{x}$ = 0.40, which indicates the presence of a pressure node. Low standard deviations of 44, 49, 54, and 51 Pa were also observed at $\bar{x}$ = 0.86, 0.66. 0.57, and 0.17, respectively. 

Figure 5b illustrates the case of 240 Hz acoustics. Unlike in 220 Hz, a prominent peak appeared only for 240 Hz at $\bar{x}$ = 0.74, whereas smaller peaks around the one at 240 Hz appeared at $\bar{x}$ = 0.66. The spread of other frequencies around the 240 Hz peak was not as wide as that around 220 Hz peak. This phenomenon can be observed across different locations where the pressure transients were recorded, and resulted in the evolution of localized high-density wave patterns as shown in Figure 4b. The highest standard deviation of 301 Pa, which indicates a pressure antinode, was recorded at $\bar{x}$ = 0.06 (near the distributor), and was almost twice that obtained for 220 Hz sound waves. The second highest standard deviation of 210 Pa was recorded much closer to the sound source at $\bar{x}$ = 0.74. 

The amplitude spectra of the pressure signals recorded at different locations when the column was excited with the 360 Hz frequency sound waves are shown in Figure 5c. The high magnitude fluctuations at $\bar{x}$ = 0.74, 0.57, 0.29, and 0.06 seen in Figure 4c can be ascribed to the formation of antinodes. However, unlike the 220 Hz and 240 Hz acoustic waves considered before, the prominent amplitude peak was almost exclusively contributed by 360 Hz acoustic disturbances. The standard deviation of the pressure fluctuations near the distributor (i.e., at $\bar{x}$ = 0.06) was 650 Pa (SPL = 146 dB), which was even higher than the 125 dB amplitude used for the sound source. At $\bar{x}$ = 0.40, the lowest degree of fluctuation was observed with a standard deviation of 16 Pa, which corresponded to an SPL of 110 dB. 

The predictions of the wave equation for the closed column at selected frequencies are shown in Figure 6, which plots the variation of the sound pressure level (SPL) with the distance above the distributor. The formation of pressure nodes and antinodes at various locations along the height of the fluidization column indicates a clear trend with a changing frequency. As the frequency increased, the nodes and antinodes also increased in number and shifted their positions. For example, the mathematical model predicted pressure nodes at $\bar{x}$ = 0.17 and at $\bar{x}$ = 0.23 above the distributor at 360 Hz (i.e., kL ≅7.2π/2), which is clearly supported by the experimental data shown in Figure 4c. Another node was observed at $\bar{x}$ = 0.69, which closely corresponds to the behavior observed at $\bar{x}$ = 0.74 (Figure 4c). A notable difference was also observed in the amplitudes of the standing waves by changing the source frequency. The amplitudes of standing waves generated with a 360 Hz source frequency is significantly higher than those of standing waves generated at lower frequency even though the experimental configuration remained the same. This behavior was also clearly reflected in the experimental data shown in Figure 3 and Figure 4.

Figure 7 compares the experimental data for 240 Hz and 360 Hz. In both cases, a clear discrepancy was observed at the lower boundary. The wave amplitude at the distributor, which was a pressure antinode, was smaller than that of the point immediately above it (i.e., $\bar{x}$ = 0.06). Such difference can be mainly attributed to the perforations on the distributor, which mitigate the development of a displacement node at the distributor. However, the model predictions and the experimental data showed an overall good agreement for the case of 360 Hz. Meanwhile, some discrepancies were noted for the case of 240 Hz due to the presence of waves of other frequencies.

### 4.2. Static Granular Nanosilica Bed Dynamics under Acoustic Vibrations

Figure 8 shows the behavior when the column is filled with nanosilica up to the height of 78 cm. Two different frequencies (i.e., 240 and 360 Hz) are considered here. Figure 8a presents the case of 240 Hz. The behavior in the freeboard region was similar to that observed for the case of an empty column as far as amplitudes are concerned. For instance, at $\bar{x}$ = 0.74, the standard deviations for the empty column and the nanosilica bed are 210.4 Pa and 210.8 Pa, respectively. Moreover, the pressure transients appeared to be superimposition of waves of different frequencies. Even the uppermost region of the bed at $\bar{x}$ = 0.40 did not show any significant difference. However, the sound intensity was significantly attenuated in the bed owing to the presence of nanosilica powder. At $\bar{x}$ = 0.06, the standard deviation was only 13.3 Pa, whereas that of empty column was 301.2 Pa. The Y-scale varies from −50 Ps to 50 Pa in the second row of the figure, whereas that in the first row varies within the ±500 Pa range.

For the same bed height, the effect of the 360 Hz acoustic waves was different than that of 240 Hz waves (Figure 8b). The amplitude of waves recorded in the freeboard region was mostly lower than those observed in the empty column even for locations close to the sound source and far from the boundary of the granular nanosilica bed (e.g., at $\bar{x}$ = 0.86). A significant difference was also observed in the freeboard region at $\bar{x}$ = 0.57, where an antinode turned into a node due to the presence of the bed. This phenomenon indicates the formation of a new standing wave pattern that evolved as a result of the interference of the reflected waves from the bed interface and the distributor. However, the magnitude of the wave had a lower resultant amplitude due to the absorption of sound energy by the granular medium as indicated by Equations (13) and (14). As seen in Figure 8b, the main difference was observed for the transients occurring inside the granular nanosilica bed. While the fluctuations for the empty bed varied by ± 600 Pa, less than ±40 Pa variations were observed in the locations inside the bed. For instance, at $\bar{x}$ = 0.06, the difference in the standard deviation was almost 100 times; 6.5 Pa (101 dB) for the nanosilica bed and 650 Pa (146 dB) for the empty column. Interestingly, the difference in the standard deviations between $\bar{x}$ = 0.40 (20.3 Pa, immediately below the top interface), and $\bar{x}$ = 0.29 (19.4 Pa, well within the bed) was negligible, although one would expect the sound waves to attenuate along with increasing penetration depth inside the bed. This phenomenon can be attributed to the evolution of a new standing wave pattern inside the porous granular nanosilica bed.

In the frequency domain (Figure 9), several peaks simultaneously appeared in the neighborhood of the dominant frequency (i.e., 240 Hz), when the system was excited with 240 Hz frequency. This phenomenon can be clearly observed in the freeboard region and the upper part of the granular bed. However, as the acoustic waves traveled down the granular bed, the other frequency components were filtered out, thereby retaining only the dominant frequency component. The bed excited with 360 Hz sound waves demonstrated similar behavior. In most cases, the dominant peak at 360 Hz was clearly visible. Although attenuated, the dominant frequency wave was retained by the granular nanosilica bed while filtering out the other frequencies.

### 4.3. Signal Power

The average power of the signal for the empty column and nanosilica bed was evaluated by integrating or summing the power spectra over the frequency. Figure 10 presents signal power only for selected frequencies (i.e., 220, 240, 360, and 380 Hz) in addition to the mean signal power of all frequencies considered in this study. Figure 10a considers the case of the empty column and clearly shows the peaks and troughs due to the existence of pressure nodes and antinodes. The sound waves of 360 Hz exhibited prominent peaks and troughs. These results were supported by the theoretical predictions in Figure 6. The dominant frequency was the main contributor to the signal power. When the column was partially filled with nanosilica, a significant change can be observed in the power profiles along the bed height as shown in Figure 10b. The nearly thousand-fold decrease in the magnitude of power inside the bed could be ascribed to the absorption of sound energy. In fact, the penetration of sound energy was limited only to the upper part of the bed. In this study, the bed height was set to 0.78 m (i.e., $\bar{x}$ = 0.45), and the first pressure port from the top of the bed was located at $\bar{x}$ = 0.40 (i.e., 0.70 m from the distributor). Despite a significant dissipation of the sound energy, the clear presence of peaks and troughs indicate the evolution of the standing wave pattern inside the bed. A pressure node was observed at $\bar{x}$ = 0.17 and an antinode was reported at $\bar{x}$ = 0.06 near the distributor. Meanwhile, the freeboard region, despite the absorption of energy in the bed, was not similarly affected as far as the power is concerned. Another noteworthy difference was observed between Figure 10a,b. The prominent peaks and troughs of 360 Hz sound waves in the empty column were suppressed due to the presence of the bed. In sum, changing the frequency of the sound waves does not significantly impact the power transmission inside the bed.

The power transmission and dissipation due to the presence of the granular nanosilica bed were further explored by comparing the power deep inside the bed at a distance of $\bar{x}$ = 0.06 from the distributor with the corresponding value for the empty column. The case of the freeboard region at $\bar{x}$ = 0.74 was also considered. The ratios for both cases are presented in Figure 11a. The ratio for the freeboard region mostly varied from 0.5 to 5. Interestingly, the average signal power (i.e., pressure fluctuations) at 300 Hz and 320 Hz in the freeboard region was almost five time larger than that in the empty column due to the shifting of the pressure antinode in response to the presence of the granular static bed. Meanwhile, the higher value of frequencies above 340 Hz resulted in significantly lower ratios. A similar trend was also observed for deep inside the bed at $\bar{x}$ = 0.06. However, the average signal power in the absence of the bed was 200 to 10,000 times higher than that in the presence of a bed, which clearly demonstrates that the presence of a granular nanosilica bed results in a significant power attenuation.

Only a single location, each for the freeboard and the bed, where the peak fluctuations occurred has been considered so far in the above. The overall average signal powers for the freeboard region extending from $\bar{x}$ = 0.57 to 0.86 and for the granular bed region extending from $\bar{x}$ = 0.0 to 0.40 were also computed. Figure 11b compares the average powers thus obtained with the corresponding values for the empty bed to gain further insights into the region-wise signal attenuations. The observed trend was similar to that shown in Figure 11a where local ratios are presented. A smaller ratio was observed at 360 Hz since the signal power at this frequency was most pronounced in the empty column. However, the presence of the granular nanosilica bed mitigated its effect, resulting in lower values of power ratios.

## 5. Conclusions

Unlike previous studies that mostly used SPL meters, sensitive pressure transduces were utilized in this work to monitor the wave patterns generated in the closed empty column and static granular bed subjected to acoustic disturbance by placing a sound source at the top of the column. The amplitudes of the signal (i.e., pressure fluctuations) differed across various locations along the height of the column, indicating the evolution of standing wave patterns. Although the sound source was excited with a single frequency, the frequency domain analysis of the signal revealed the presence of disturbances of other frequencies. However, the granular nanosilica bed filtered the disturbances and retained only the dominant frequency component of the signal. A careful comparison of the disturbances between the empty column and the static granular bed revealed a significant dissipation of the signal power as the waves propagated downward the bed toward the distributor. The effect of the sound waves could only be observed in the upper region of the bed. Compared with the empty column, the signal power decreased by several orders of magnitude along with increasing distance from the upper interface of the bed. The sound frequency clearly affected the signal strength. However, contrary to the case of the empty column, the locations of pressure nodes and antinodes were not affected by the change of the frequency in the nanosilica bed.

## Figures and Tables

**Figure 1 nanomaterials-12-01509-f001:**
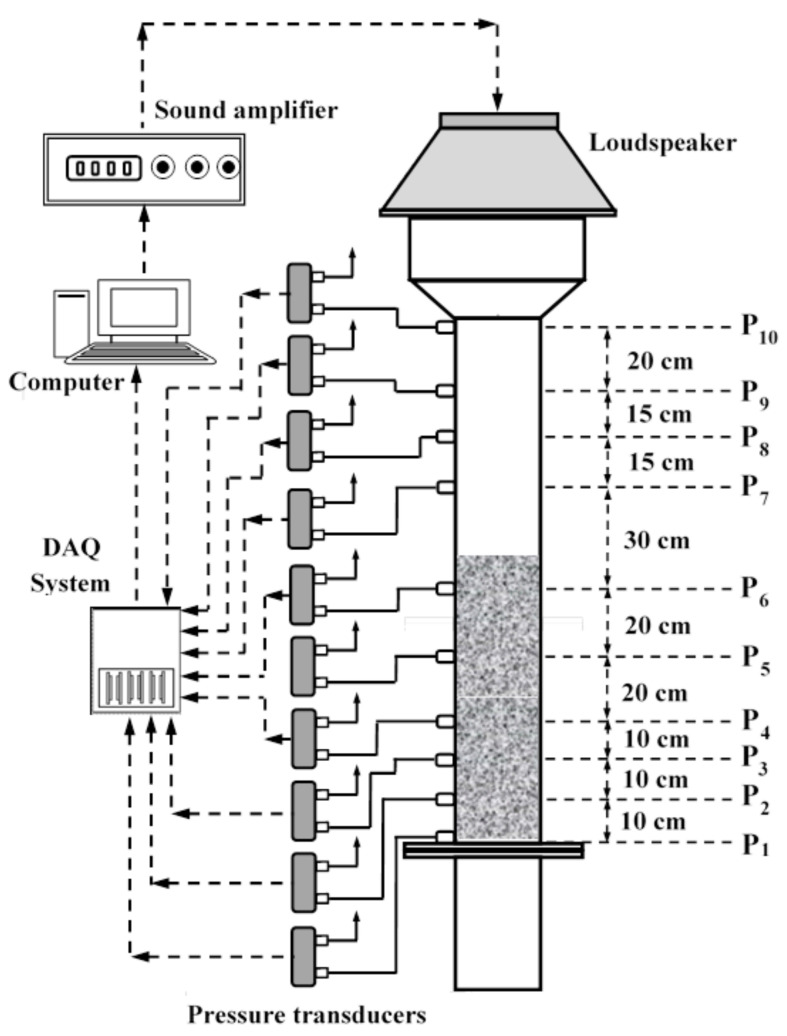
Schematic of the experimental set-up.

**Figure 2 nanomaterials-12-01509-f002:**
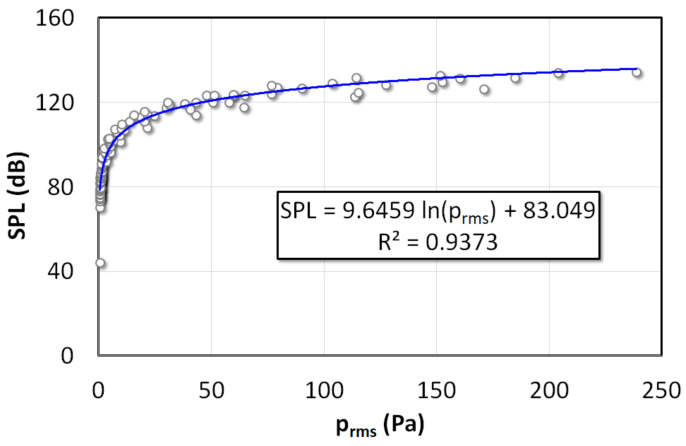
Calibration of the SPL meter with respect to root mean square pressure amplitude.

**Figure 3 nanomaterials-12-01509-f003:**
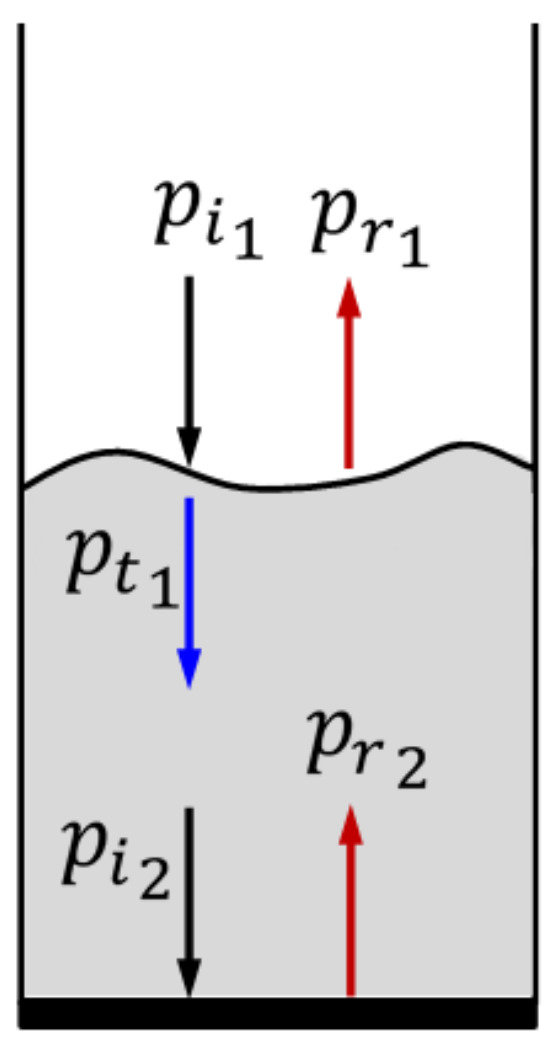
Incidence, reflection and transmission of acoustic waves from the bed interface and the distributor.

**Figure 4 nanomaterials-12-01509-f004:**
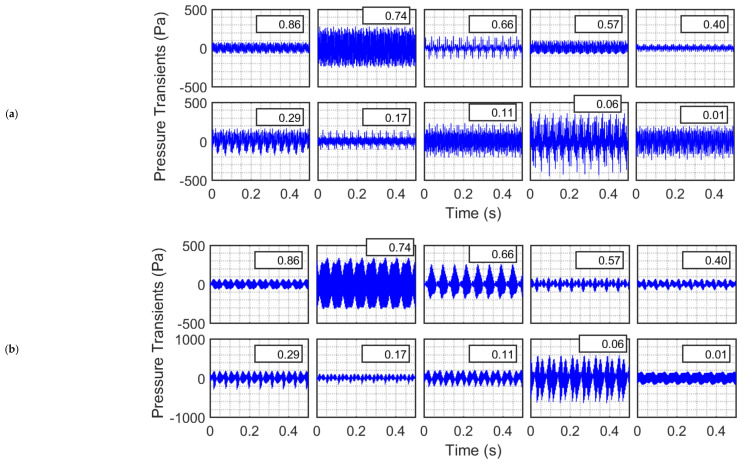
Local pressure transients in a closed empty column excited with sound waves of frequencies (**a**) 220, (**b**) 240, and (**c**) 360 Hz.

**Figure 5 nanomaterials-12-01509-f005:**
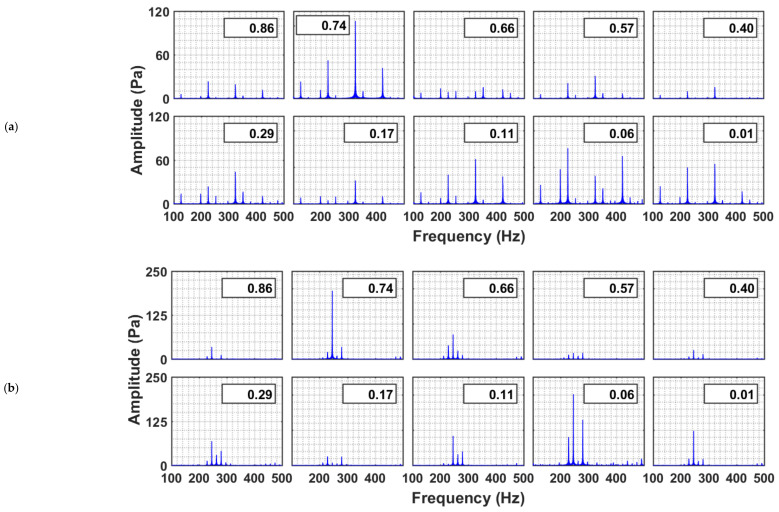
Frequency spectra of the local pressure transients in a closed empty column subjected to sound waves of frequencies (**a**) 220, (**b**) 240, and (**c**) 360 Hz.

**Figure 6 nanomaterials-12-01509-f006:**
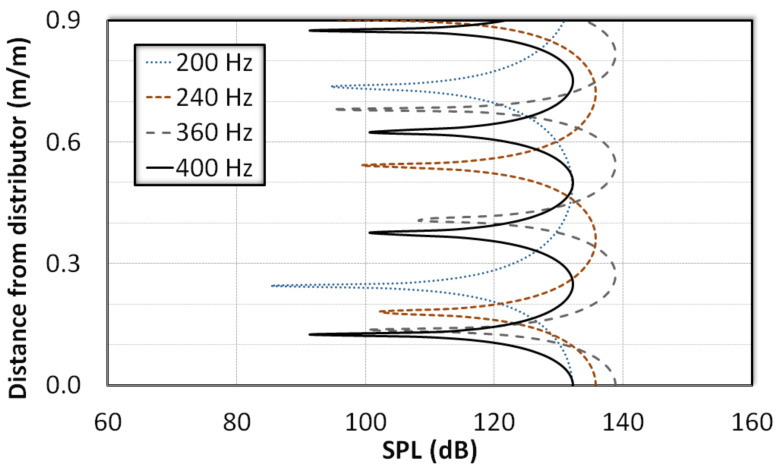
Model predictions of sound pressure level in an empty column subjected to sound waves of different frequencies.

**Figure 7 nanomaterials-12-01509-f007:**
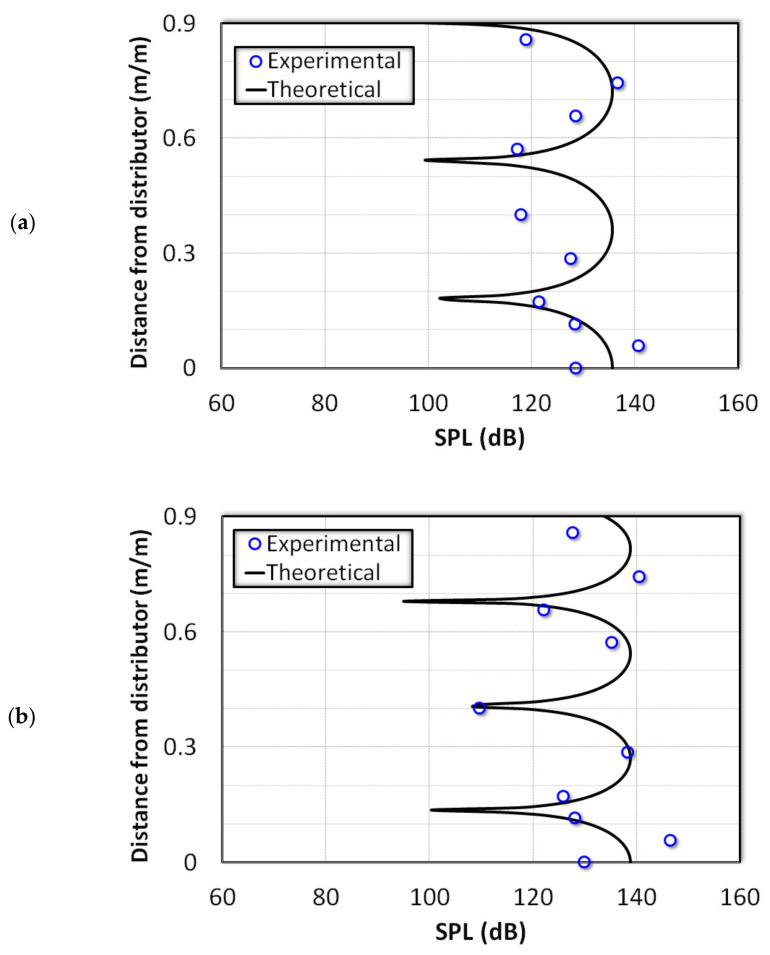
Comparison of model predictions with experimental data in an empty column subjected to sound waves of frequencies (**a**) 240 Hz and (**b**) 360 Hz.

**Figure 8 nanomaterials-12-01509-f008:**
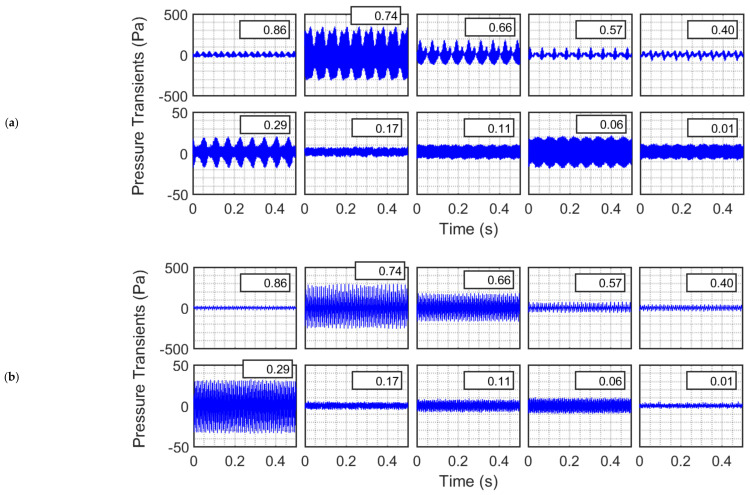
Pressure transients in a nanosilica bed and freeboard region excited with sound waves of frequencies (**a**) 240 Hz and (**b**) 360 Hz.

**Figure 9 nanomaterials-12-01509-f009:**
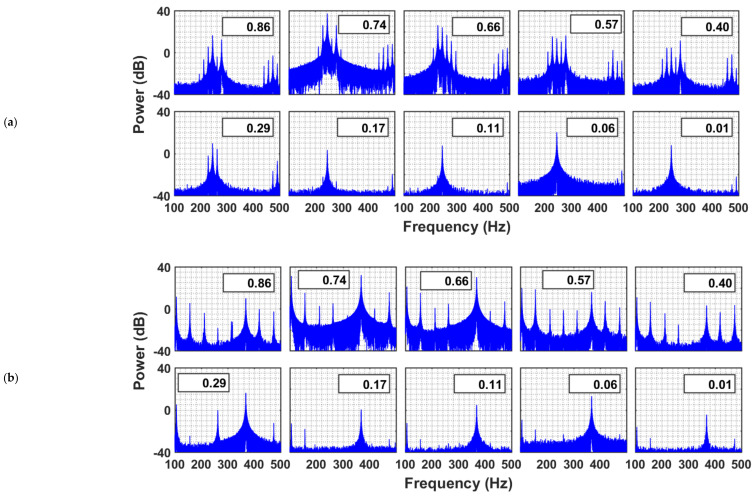
Power spectra of pressure transients in a nanosilica bed and freeboard region excited with sound waves of frequencies (**a**) 240 Hz and (**b**) 360 Hz.

**Figure 10 nanomaterials-12-01509-f010:**
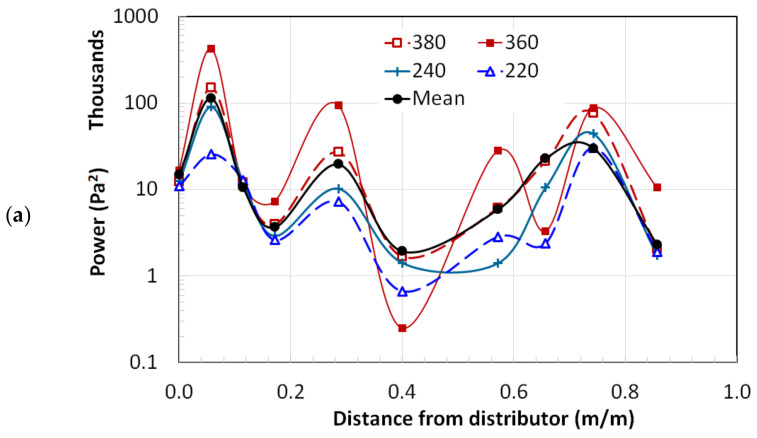
Power distribution of sound waves along the column length for (**a**) empty column and (**b**) nanosilica bed.

**Figure 11 nanomaterials-12-01509-f011:**
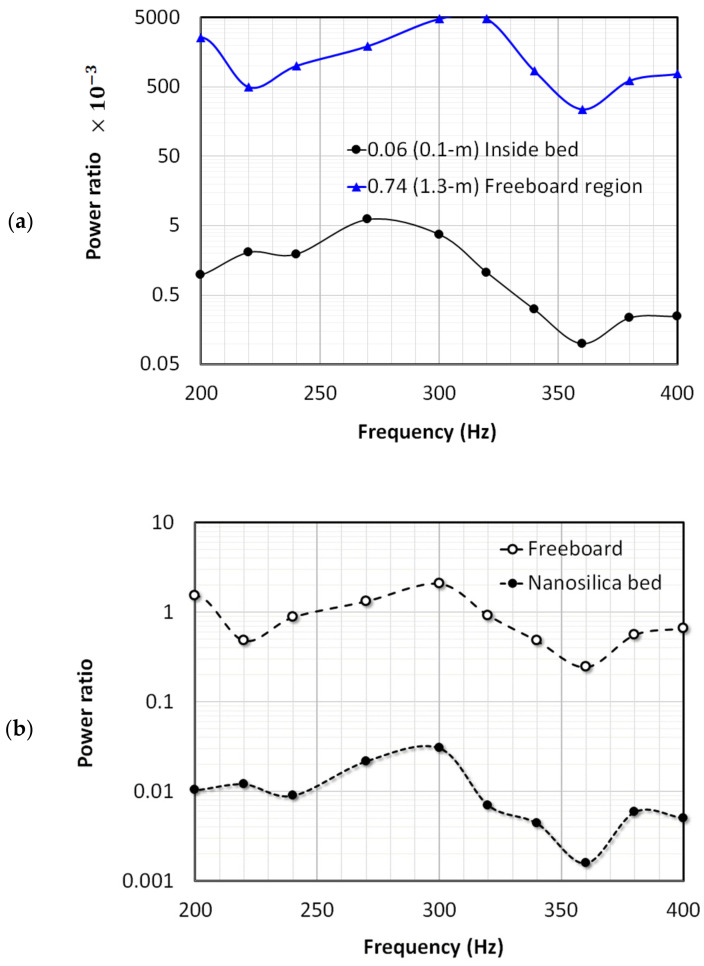
Effect of frequency on the acoustic attenuation (**a**) at a specific point inside the bed and in the freeboard region, and (**b**) the average effect for the entire bed and the freeboard region (the empty column values are used as reference).

**Table 1 nanomaterials-12-01509-t001:** Location of pressure ports with distances measured from the distributor.

Pressure Port	P1	P2	P3	P4	P5	P6	P7	P8	P9	P10
Location, *x* (m)	0.01	0.1	0.2	0.3	0.5	0.7	1.0	1.15	1.3	1.5
Location $\bar{x}$ (m/m)	0.01	0.06	0.11	0.17	0.29	0.40	0.57	0.66	0.74	0.86

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
