# Peer review of "A Detailed Insight into Acoustic Attenuation in a Static Bed of Hydrophilic Nanosilica"

_nanomaterials, 2022, doi:10.3390/nano12091509_

Round 1

Reviewer 1 Report

Title

Advisable to change to “The effectiveness of acoustic attenuation in a static bed of hydrophilic nanosilica as energy transport”

Abstract

The abstract is confusing and not focus.

The authors need to emphasize and highlight the aim of this study, the methodology and highlight the significant performance that reflect the title the manuscript. Also need to mention the novelty of this study at the end of the abstract.

Introduction

Sufficient. However, please mention your objective in one separate paragraph for clear and easy reading.

Experimental

3.1. Subsection ? please change into ***?

3.1 or 2.1 ?

Line 129: The detailed schematic of the setup is reported in Figure 1 --? What setup ?

Figure 3. This is a figure. Schemes follow the same formatting. ? where is Figure 2. Then, what is really caption for this Figure ??

Results and discussion

Sufficient. However, authors need to rearrange the paragraph neatly.

Conclusions

Sufficient.  

Reviewer 2 Report

Interesting experiment was presented. Comments for manuscript enhancement:

Introduction.

Different nanomaterials have been studied in acoustic fields, and authors may refer to a recently published review article on nanoacoustics (Peng, et al., Acoustics at the nanoscale (nanoacoustics): A comprehensive literature review. Sensors and Actuators A: Physical, 2021) and point out the uniqueness of this paper.

Page 1, Line 30: Typo. Correct to: “due to the”.

Page 2, Lines 68-69: “…for group C particles, while remained relatively unaffected for group A particles…”. The second half of the sentence is not clear and there appears to be a word missing “while [X] remained”.

Experimental.

Page 3, Line 144: the sensitivity and bandwidth of the transducers and the rationale of using this transducer may be added here.

2 equations on pages 4 and 5 were not numbered.

Most of the acoustic propagation equations are well known and the derivations are necessary.

Page 5, Line 182: Why was this frequency range chosen? Why are the increments not consistently 20 Hz but there is a brief 30 Hz jump at 240 Hz and 270 Hz?

Mathematical model

Page 7, Line 222: Typo: “that that” repeated word.

Page 9, Line 250: Clarity: “because of attenuation of acoustic due”. Seems to be missing a word between “acoustic” and “due”.

Page 9, Line 253: Need to adjust description for the Figure 3 label.

Results and discussion

Page 9, Lines 271 and 276: Typos, replace with “dynamics are significantly different” and “decreases until at x”.

Page 11: Page numbers reset on the manuscript pages and start from 1.

Page 15, Line 350: Where is the data from 220 Hz? We have compared 220, 240, and 360 Hz up until now. Why is it absent?

Page 17, Line 387: Sentence is not clear. “is nonetheless for the transients”.

Page 21, Line 445: Fix the x_bar symbol.

Difference between the previous SPL meter and the current pressure transducer results can not be found.

Reviewer 3 Report

In the article “A detailed insight into acoustic attenuation in a static bed of hydrophilic nanosilica” local dynamics of a nano-silica bed, excited with sinusoidal acoustic waves of different frequencies, were carefully monitored. The experimental method is reasonable and the results are convincing. It is recommended to be accepted after minor revision.

  1. The conclusions in the abstract lack of specific support.
  2. The significance of this work is not clarified in the introduction.
  3. The legend symbols in figures 8-10 are not marked clarely.
  4. Language should be polished.

Round 2

Reviewer 2 Report

previous review comments were well responded.